# Exploring the Importance of Environmental Complexity for Newly Hatched Zebrafish

**DOI:** 10.3390/ani14071031

**Published:** 2024-03-28

**Authors:** Maria Santacà, Elia Gatto, Marco Dadda, Matteo Bruzzone, Marco Dal Maschio, Angelo Bisazza

**Affiliations:** 1Department of Behavioral and Cognitive Biology, University of Vienna, 1030 Vienna, Austria; 2Department of General Psychology, University of Padova, 35131 Padova, Italy; marco.dadda@unipd.it (M.D.);; 3Department of Chemical, Pharmaceutical and Agricultural Science, University of Ferrara, 44121 Ferrara, Italy; elia.gatto@unife.it; 4Department of Life Science and Biotechnology, University of Ferrara, 44121 Ferrara, Italy; 5Padua Neuroscience Center, University of Padova, 35131 Padova, Italymarco.dalmaschio@unipd.it (M.D.M.); 6Department of Biomedical Sciences, University of Padova, 35131 Padova, Italy

**Keywords:** animal behaviour, animal cognition, developmental plasticity, *Danio rerio*, environmental complexity

## Abstract

**Simple Summary:**

Growing up in an impoverished environment can profoundly affect brain development and cognitive abilities in animals. Most studies have looked at the effects on adults and we know little about how it impacts the early stages of cognitive development in fish. We studied zebrafish larvae, which are routinely reared in bare containers where they experience reduced sensory stimulation. We aimed to see if providing them a more enriched environment would have an effect on their cognitive abilities and behaviour. Larval zebrafish from the enriched treatment had better performance than controls in a number discrimination task but did not differ when required to discriminate two areas. In both experiments, larvae from the impoverished treatment showed a reduced locomotor activity. When essayed in a preference test, recently hatched larvae showed a marked preference for an enriched environment over an empty one. A better understanding of these effects is crucial for the welfare of captive zebrafish populations as well as for the quality and reliability of research on larval zebrafish.

**Abstract:**

The effects of an early impoverished social or physical environment on vertebrate neural development and cognition has been known for decades. While existing studies have focused on the long-term effects, measuring adult cognitive phenotypes, studies on the effects of environmental complexity on the early stages of development are lacking. Zebrafish (*Danio rerio*) hatchlings are assumed to have minimal interaction with their environment and are routinely reared in small, bare containers. To investigate the effects of being raised under such conditions on development of behaviour and cognition, hatchlings housed for 10 days in either an enriched or a standard environment underwent two cognitive tasks. The results were mixed. Subjects of the two treatments did not differ in performance when required to discriminate two areas. Conversely, we found a significant effect in a number discrimination task, with subjects from impoverished condition performing significantly worse. In both experiments, larvae reared in impoverished environment showed a reduced locomotor activity. Given the effects that enrichment appears to exert on larvae, a third experiment explored whether hatchlings exhibit a spontaneous preference for more complex environments. When offered a choice between a bare setting and one with objects of different shapes and colors, larvae spent over 70% of time in the enriched sector. Deepening these effects of an early impoverished environment on cognitive development is crucial for the welfare of captive zebrafish populations and for enhancing the quality and reliability of studies on larval zebrafish.

## 1. Introduction

Building on pioneering studies from the 1960s, it is well-established that rearing mammals in impoverished environments can exert profound and irreversible effects on neural development, significantly impacting the development of several cognition functions, including learning, memory, and visual discrimination [1,2,3]. This phenomenon extends beyond mammals, with subsequent research documenting similar effects of early impoverished social or physical environments across various vertebrate taxa (reviewed in [4,5,6]).

In teleosts, specifically, studies conducted on captive species have elucidated the positive outcomes of environmental enrichment. For example, environmental enrichment has been found to promote neural plasticity and behavioural flexibility in Atlantic salmon (*Salmo salar*; [7]), enhance social skills in rainbow trout (*Oncorhynchus mykiss*; [8]), foster social learning and antipredator behaviour in cod (*Gadus morhua*; [9,10]), and positively influence the development of spatial abilities in rainbow trout [11].

While much research on fish has focused on the long-term effects of environmental complexity, particularly in measuring cognitive phenotypes in adults, there is a notable gap in understanding the effects of environmental complexity on behaviour and cognition during the early stages of development. Unlike warm-blooded vertebrates, oviparous teleosts are orders of magnitude smaller than an adult at birth, typically emerging at a rather undeveloped stage, and research has been limited by the difficulty to develop tools suitable for investigating cognitive functions at this age.

Over the last three decades, increasing attention has been directed toward the welfare of animals kept in captivity, prompting substantial research on the potential development of enrichment programs to enhance the conditions for farm, companion, and captive exotic animals (e.g., [12,13,14,15]. Recently, equal emphasis has been given to the quality of the rearing environment for species commonly used in scientific research. Laboratory species, typically raised in standard environments, are now recognized as benefiting from structural complexity or rich social environments, leading to enhanced overall well-being in birds, rodents, lagomorphs, and teleosts [16,17,18,19].

The significance of keeping laboratory animals in adequate environmental conditions extends beyond the welfare considerations of laboratory animals; it also holds implications for the quality and reliability of research outcomes. Enriched environments have the potential to yield more robust and reproducible experimental results, as stressed or unhealthy animals may not respond predictably to experimental treatments [20,21,22].

The zebrafish (*Danio rerio*) is the most widely used teleost in scientific research and is overall one of the emerging models in translational research across a wide variety of fields. Several studies have presented compelling evidence that environmental complexity improves the welfare and affects behaviour and cognition of adult zebrafish (reviewed in [23,24]). Physical enrichment has been incorporated into guidelines for the housing and care of fish used for scientific purposes. However, these guidelines often lack specific recommendations for zebrafish, and, more critically, there is a notable absence of guidance on the specific types of physical enrichment needed for zebrafish at various developmental stages [25]. Due to the hatching of eggs occurring only 48–72 h after fertilization and the larvae being born at a highly immature stage of development [26], a longstanding assumption has prevailed that, in the first few days of life, zebrafish have limited interaction with the surrounding physical world. They are typically kept in small, bare containers such as Petri dishes, with the result that, during their first weeks of life, they grow in an exceptionally impoverished environment in which visual and olfactory stimulation is reduced to a minimum [27,28].

At the same time, the fact that newly hatched larvae exhibit a rather immature state of development and a very limited behavioural repertoire, considerably restricted the possibility of investigating this issue. The only exception to this scarcity can be found in two studies that, using a very simple behavioural response, evidenced a reduction in neophobic responses among larval zebrafish when reared in an enriched environment [29,30]. However, new tools are developing to study the cognitive functions of larval zebrafish. For example, by immobilizing 7–8 days post fertilization (dpf) larvae in agarose, it is possible to demonstrate their susceptibility to conditioning through repeated pairing of a moving spot of light with a touch of their tail [31]. Moreover, a recent study showed that larval zebrafish learn in five days to discriminate two objects that differ for colour or shape by repeatedly pairing one stimulus with food [32]. Both methods employ complex procedures, involve a series of larval manipulations, and extend over long periods. This makes them poorly suited for the purpose as the treatments required for measurement may obscure subtle differences resulting from having reared them in different environments. Two other methods have recently been developed, based on spontaneous preferences, that allow for relatively quick measurement of two cognitive abilities. The first method exploits an innate tendency of zebrafish to navigate an obstacle by passing through the wider hole when more options are available. In a spontaneous choice, both larval and adult zebrafish were found to be extremely accurate in this task being able to discriminate two holes differing by less than 10% in area [33,34]. The second method stems from the observation that zebrafish, like other animals, show a preference for environments with vertical stripes, likely because they simulate vegetated habitats [35,36,37]. A study conducted on 7 dpf larvae, showed that in a spontaneous choice test they choose the more numerous of two arrays of stripes, suggesting a numerical discrimination capacity [38].

If zebrafish postnatal nervous system development follows a pattern similar to that of birds and mammals, the early environment’s characteristics are expected to significantly impact their sensory system, behaviour, and cognition. To explore this, we raised freshly hatched larvae until 13 dpf in two distinct environments: one enriched with various objects differing in colour and shape, and the other in the traditional impoverished laboratory conditions. After the treatment period, we compared the behaviour and cognitive abilities of the two groups. We utilized the two tests described earlier to measure numerical abilities and continuous quantities discrimination [33,38]. Since we obtained some evidence that rearing zebrafish in a complex environment positively affects their cognitive development, our third experiment investigated whether larvae exhibit a spontaneous preference at birth for an environment with those characteristics.

## 2. Materials and Methods

### 2.1. Animal Housing

Subjects were wild-type zebrafish larvae obtained from many different spawning of adult breeders from a wildtype strain bought by a local supplier in 2018 and maintained in the facility of the Department of General Psychology of University of Padova (Italy). At hatching larvae tend to remain inactive, lying on one side on the bottom of the tank. Free swimming starts at 4–5 dpf, after the inflation of the bladder. Until 5 dpf, larvae were maintained in Petri dishes (10 cm Ø, h: 1.5 cm; approx. 50 individuals per dish) in a solution of Fish Water 1× (for the detailed concentration see [39]) and Methylene blue (0.0016 g/L). Methylene blue is an antifungal agent which increases larval survival, and it has no effect on fish development and behaviour [40]. Petri dishes were all maintained in the same room at a temperature of 28.5 ± 1 °C and lit according to a 14:10 h light:dark cycle. Larvae were fed twice a day with dry food (an admixture of GEMMA Micro 75 and TetraMin flakes, particle size: 0.75 mm) from the age of 6 dpf. To prevent any negative effect on explorative activities of larval zebrafish, all experimental phases (i.e., conditioning treatment in Experiment 1 and 2, and test phase for the three experiments; see details below) were conducted in the same dark room exclusively designated to perform the present study. All conditioning and testing tanks were inserted into large white plastic tank (160 cm × 90 cm, height: 60 cm) placed on a table and were lit by two 0.72-W strip LED lamps. Room conditions (e.g., air and temperature) and water parameters were monitored daily during the prolonged experimental period [41].


**Experiment 1: Continuous quantity discrimination**


The procedure of the first experiment is the same used by a recent study on the ontogeny of the ability to discriminate quantities in zebrafish larvae [33]. Previous studies have reported that zebrafish larvae display social behaviour starting between 10 and 16 dpf (e.g., [42,43,44]), suggesting that being in a group from the early-stage of life is crucial for behavioural and cognitive development. Since our experimental procedure lasted four days, larvae were tested in groups of six individuals and each group was considered as one data point (see “statistical analysis” section). We tested 96 zebrafish larvae (*n* = 48 larvae per treatment) for a total of 16 groups of 6 larvae each.

**Pre-experimental condition: Environmental enrichment and standard treatments.** At 5 dpf, we gently transferred a group of 30 larvae from Petri dishes to treatment tanks, i.e., 3D-printed rectangular boxes (14 cm × 7 cm × 4 cm filled to 3 cm) with white PLA and filled with Fish Water 1× diluted with Methylene blue solution (0.0005 g/L). Larvae were randomly split either in one of two treatment tanks. The environmental enriched (EE) treatment tank contained 8 Lego^®^ objects of various colour and shape. At 8 and 12 dpf, two pieces were replaced with a pair of different shape and colour to increase the stimulation and to maintain the novelty of the environment [45]. The Lego^®^ bricks were fully covered of water, thus larvae could visually and tactility interact with objects, and they were placed in all possible positions and orientations (i.e., vertically versus horizontally). For the standard treatment (ST) we used a bare tank that contained no object. Larvae were maintained in each treatment until the start of the cognitive tests and fed with dry food (particle size: 0.75 mm) twice per day.

**Apparatus and test procedure.** The apparatus consisted of an hourglass-shaped apparatus (12 cm × 4.8 cm × 4 cm) 3D-printed with white PLA and filled with 3.5 cm of Fish Water 1× (Figure 1). A central corridor (length: 4.3 cm) divided the apparatus into a frontal and a posterior sector. In the middle of the corridor, larvae were presented with a panel of grey PLA material (3 cm × 3.2 cm; Figure 1). The panel presented two holes through which larvae could spontaneously pass to move from the two sectors of the apparatus. We used four identical apparatuses at the same time and one 0.72-W strip LED lamp was placed 1 cm above each sector of each apparatus. One camera (Canon LEGRIA HF R38, Canon Inc., Tokyo, Japan) was placed 90 cm above the central corridors to record passages.

The experimental procedure consisted of a familiarisation phase followed by the test phase. Larvae started the familiarisation phase at 14 dpf and were 16 and 17 dpf at the time of the test phase. On the first day of the familiarisation phase, we randomly selected six larvae from one treatment tank (EE or ST) and inserted them in one experimental apparatus. No panel was present on the first day. On the second day, we inserted a grey panel with a single central hole (0.7 cm in diameter) so to accustom larvae to pass through the hole to move from one sector to the other. We video recorded the second day of the familiarisation phase to ensure that the larvae had learned to move between sectors.

In the test phase, we presented four panels with four different size discriminations. The values of the area ratio between the smaller and the larger hole ratios were 0.60, 0.75, 0.86, and 0.91 (Figure 1; Appendix A). In each day, we observed groups for eight consecutive hours during which we presented all four ratios for a 2 h period each. Presentation order of the four ratios and left–right position of the larger hole were randomized across groups.

We analysed the recordings offline, played back on a computer. In particular, we scored the total number of passages through the larger or smaller hole.


**Experiment 2: number discrimination**


The procedure of the second experiment is similar to that used by a recent study on the ontogeny numerical abilities in zebrafish [38]. Overall, we tested 40 zebrafish larvae (*n* = 20 larvae per treatment). Twenty subjects were tested in a 2 vs. 4 number discrimination (*n* = 10 larvae per treatment); the remaining 20 subjects were tested in a 3 vs. 4 number discrimination (*n* = 10 larvae per treatment).

**Pre-experimental condition: Environmental enrichment and standard treatments.** As in the previous experiment, at 5 dpf subjects were randomly split into two groups and one half was assigned to the environmental enriched (EE) treatment, the other half to the standard treatment (ST). All other details are the same described in Experiment 1.

**Apparatus and test procedure.** The test apparatus (Figure 2) consisted of a tank similar to that used for the pre-experimental treatments but of a smaller size (7 cm × 4 cm × 4 cm). The apparatus was filled with 2.5 cm of Fish Water 1× and placed in a room kept at 28.5 ± 1 °C and illuminated by two 0.72-W strip LED lamps placed 90 cm above each sector of each apparatus. The two visual stimuli to be discriminated were placed along the two short walls of the apparatus (Figure 2). The stimuli were two-dimensional figures made with Microsoft PowerPoint (Microsoft 365 MSO; Version 2210) and laser printed on 4 cm × 5 cm white laminated card. A camera (Canon LEGRIA HF R38, Canon Inc., Tokyo, Japan) positioned at 90 cm above the apparatus recorded the test. 

At the beginning of the test (14 dpf), one subject was collected with a plastic Pasteur pipette and gently transferred into the centre of the apparatus and left free to move in the apparatus and to interact with the stimuli for 12 min (testing time). The dimensions of the apparatus allowed subjects to simultaneously observe the two stimuli. In the 2 vs. 4 number discrimination, one stimulus consisted of two 0.3 cm × 3 cm vertical bars (distance between bars: 0.6 cm; cumulative surface area: 1.800 cm^2^; Figure 2), the other of four 0.3 cm × 3 cm vertical bars (distance: 0.6 cm; area: 3.600 cm^2^; Figure 2). Instead, in 2 vs. 3 number discrimination, one stimulus consisted of two 0.3 cm × 3 cm vertical bars (distance: 0.6 cm; area: 1.800 cm^2^; Figure 2), the other of three 0.3 cm × 3 cm vertical bars (distance 0.6 cm; area: 2.700 cm^2^; Figure 2). All the bars were black and presented on a white background. The position of the stimuli was counterbalanced across subjects. The stimuli were only controlled for stimulus density and no other continuous variable.

We analysed the recordings offline, played back on a computer. The apparatus was virtually subdivided into the three rectangular 4 cm × 2.33 cm sectors. We measured the time spent in the two sectors placed close to each stimulus. The third, central sector was considered to be a no-choice sector. The position of the subject was determined thought a custom written tracking written in Python (see [29,30] for details). For each subject, the software provided velocity, distance travelled and position, then calculating the percentage of time spent in the two choice areas.


**Experiment 3: Preference for enriched environment**


**Apparatus.** In the third experiment, we used six identical rectangular (7 cm × 14 cm, height: 4 cm) apparatuses 3D-printed with white non-toxic polylactic acid (PLA). Each tank was filled with 3.5 cm of Fish Water 1× diluted with Methylene blue solution (0.0005 g/L). In one half of the tank (i.e., the enriched sector), we presented 5 Lego^®^ objects of various colours (Figure 3) that larvae could visually and tactility interact with, while the remaining part of the tank was maintained empty (Figure 3). The half of the tank containing the enrichment was randomly chosen and counterbalanced among tanks. The experiment was videorecorded using two cameras (Canon LEGRIA HF R38, Canon Inc., Tokyo, Japan) placed 60 cm above the table.

**Procedure.** Subjects were maintained in a Petri dish until they reached the 5 dpf when the experiment begun at 17.00 pm. Larvae were gently moved to the experimental apparatus using a Pasteur pipette. Twenty larvae were transferred to each tank by releasing them in the middle of the apparatus (120 larvae in total).

The six tanks remained undisturbed for the following 24 h. Larvae were not fed for the duration of the experiment (duration: 2 days; larvae age at the end of test: 6 dpf). Although zebrafish larvae begin feeding as early as 5 dpf, food supply is not crucial for survival and growth until 8 dpf [46]. We video recorded the behaviour of the fish in each tank for 90 min soon after their introduction (trial 1: 17.00 to 18.30 pm), for 60 min the following day in the morning (trial 2: 10.00 to 11.00 am) and for 60 min in the afternoon (trial 3: 16.00 to 17.00 pm).

We analysed the recordings offline, played back on a computer. The tank was subdivided into two virtual sectors and an experimenter blind with respect to the research aim manually recorded the number of larvae in each half of the apparatus at 30 s intervals by slowing down the video recording at 0.25× to increase accuracy. Preference for the enrichment was calculated as the proportion of larvae presented in the enriched part of the tank/the total number of larvae.

### 2.2. Statistical Analyses

Analyses were performed in R version 4.3.2 (The R Foundation for Statistical Computing, Vienna, Austria, http://www.r-project.org, accessed on 2 December 2023). The significance threshold was set at *p =* 0.05.

**Experiment 1.** Each group of six larvae was considered as one datapoint with no distinction between the six larvae (total sample size = 256). To evaluate whether enrichment has an effect also on larvae behaviour, we performed a linear mixed-effects model (LMM, “lme” function from the “nlme” R package) to compare the total number of passages between the four ratios for both treatments. To do so, we fitted the LMM with treatment and ratio as fixed effects, and larvae group as random effect to handle the repeated measures design of the within-individual observation.

To evaluate whether enrichment has an effect on the cognitive abilities underlying number discrimination, we performed binomial tests to compare the passages through the larger hole for each ratio. Additionally, to compare discrimination performance (passages through the larger hole) between the different ratios, between the two treatments and the effect of the day, we used a LMM fitted with larvae group as random effect and with treatment, day, and ratio as fixed effects. When finding significant interaction between those fixed factors, we further investigated it by performing all pairwise comparisons with Tukey post-hoc tests.

**Experiment 2.** To evaluate whether enrichment has an effect also on larvae’ behaviour, we analysed the velocity and the distance covered in all sectors (both after log transformation due to right-skewed distribution) by using LMMs fitted with the numerosity (smaller or larger) and the treatment as fixed factors, as well as the subject ID as random factor.

To evaluate whether enrichment has an effect on the cognitive abilities underlying number discrimination, we analysed subjects’ time spent in each choice sector (log-transformed) by using a LMM. Analysis of time in the two choice sectors allow to determine preference for one of the two stimuli: in case of a significant preference for a certain stimulus, subjects were expected to spend more time close to that stimulus; the contrary was expected in case of avoidance of a certain stimulus. We fitted the LMM with the numerosity (smaller or larger), the number discrimination (2 vs. 3 or 2 vs. 4) and the treatment as fixed factors, as well as the subject ID as random factor. Tukey post hoc testes were used to investigate significant effect of interaction.

**Experiment 3.** To evaluate whether zebrafish larvae showed a preference for the enrichment, we firstly analysed the overall proportion of larvae presented in the enriched half of the tank in the first trial (day 1, afternoon) by using a one-sample *t*-test against chance level (0.5). The explorative tendency during the test was further analysed using a LMM fitted with block-interval (15 min) as covariate and group ID as random factor.

A similar approach was used to evaluate this preference during the second (day 2, morning) and third (day 2, afternoon) trials. Overall preference for the enrichment was assessed via one-sample *t*-test against chance level (0.5). Changes in preferences across the three trials were analysed with a LMM fitted with trial as covariate and group ID as random factor.

## 3. Results


**Experiment 1: size discrimination**


**Behavioural differences.** The mean number of passages was 817.13 ± 114.67 in 16 h of recordings for the enriched treatment larvae and 635.75 ± 120.29 for the control treatment larvae. The two treatments did not significantly differ (LMM: *F*_(1, 14)_ = 0.245, *p* = 0.629; Figure 4); there was no effect of the ratio (*F*_(3, 99)_ = 0.812, *p* = 0.490) and the interaction treatment x ratio was not significant (*F*_(3, 99)_ = 0.104, *p* = 0.958).

Since this experiment had a long familiarisation phase (two days) it is possible that behavioural differences were present at the beginning of the experiment and then subsided during the test phase. To check this possibility, we analysed the number of passages through the central hole during the familiarisation phase. The mean number of passages was 198.54 ± 57.39 in 8 h of recordings for the enriched treatment larvae and 74.58 ± 41.51 for the control treatment larvae. The difference between the two treatments is significant (LMM: *F*_(1, 11)_ = 24.497, *p* < 0.001; Figure 4).

**Cognitive differences.** Larvae of both the enriched and the control treatment passed significantly more through the larger hole in all the ratios presented (Figure 5; Appendix A). Discrimination performance differed between the four ratios (LMM: *F*_(3, 94)_ = 9.752, *p* < 0.001; linear trend: Estimate −0.121 ± 0.024, *t*_(103)_ = −5.009, *p* < 0.001). The 0.60 ratio differed from the other three ratios (Tukey post hoc test; all *p*-values < 0.01) whereas the other ratios did not differ between them (all *p*-values > 0.534). The effects of the treatment and of the day were not significant (treatment: *F*_(1, 13)_ = 0.343, *p* = 0.568; day: *F*_(1, 94)_ = 1.248, *p* = 0.267). All the interactions were not significant either (all *p*-values > 0.529).


**Experiment 2: number discrimination**


**Behavioural differences.** In the 2 vs. 4 number discrimination, enriched larvae swam significantly faster than control larvae (Enriched larvae: 6.69 ± 1.32 mm/s, mean ± SD; Control larvae: 5.15 ± 1.85 mm/s; LMM: *F*_(1, 18)_ = 4.887, *p* = 0.040) regardless of the sector (*F*_(1, 38)_ = 3.382, *p* = 0.060; interaction sector × experience treatment: *F*_(1, 38)_ = 0.019, *p* = 0.892). The results are similar in the 2 vs. 3 discrimination (speed= Enriched larvae: 6.67 ± 0.64 mm/s, Control larvae: 5.22 ± 1.44 mm/s, *F*_(1, 18)_ = 7.943, *p* = 0.012; sector: *F*_(1, 38)_ = 3.760, *p* = 0.060; interaction: *F*_(1, 38)_ = 0.182, *p* = 0.672). When considering the distance covered in the 2 vs. 4 number discrimination, enriched larvae travelled a greater distance compared to control larvae (Enriched larvae: 1721.23 ± 330.99 mm; Control larvae: 1215.59 ± 526.64 mm; LMM: *F*_(1, 18)_ = 5.034, *p* = 0.038) regardless of the sector (*F*_(1, 38)_ = 1.860, *p* = 0.170; interaction sector × experience treatment: *F*_(1, 38)_ = 2.122, *p* = 0.153). The results are similar in the 2 vs. 3 discrimination (Distance= Enriched larvae: 1685.31 ± 257.66 mm; Control larvae: 1123.572 ± 365.69 mm; *F*_(1, 18)_ = 8.652, *p* = 0.009; sector (*F*_(1, 38)_ = 1.018, *p* = 0.372; interaction: *F*_(1, 38)_ = 3.198, *p* = 0.053).

**Cognitive differences.** The total time spent cumulatively in the two choice areas did not differ between larvae of the two treatments (LMM: *F*_(1, 72)_ = 3.911, *p* = 0.052). Overall subjects spent significantly more time in the sector with the larger number of bars (larger: 253.15 ± 86.58 s, mean ± SD; smaller: 211.93 ± 84.92 s; *F*_(1, 72)_ = 4.643, *p* = 0.035) with no difference between the two number discriminations (*F*_(1, 72)_ = 0.002, *p* = 0.965). The interaction numerosity × discrimination × treatment was significant (*F*_(1, 72)_ = 5.704, *p* = 0.020). No other interaction was significant (all *p*-values > 0.175). To investigate the three-ways interaction, we conducted separate analyses for the two numerical tasks. In the 2 vs. 4 discrimination, larvae spent more time in the sector with the larger number of bars (numerosity: *F*_(1, 36)_ = 5.169, *p* = 0.029; treatment: *F*_(1, 36)_ = 0.825, *p* = 0.370), with a significant numerosity × treatment interaction (*F*_(1, 36)_ = 4.411, *p* = 0.042; see Appendix A). A Tukey post hoc test revealed that only enriched larvae discriminated the numerosities (enriched larvae: *p =* 0.006; control larvae: *p* = 0.904). In the 2 vs. 3 discrimination, larvae did not differ in the time spent in the two sectors (numerosity: *F*_(1, 36)_ = 4.088, *p* = 0.051; treatment: *F*_(1, 36)_ = 0.379, *p* = 0.542; numerosity × treatment interaction *F*_(1, 36)_ = 1.443, *p* = 0.238; Figure 6) indicating that neither enriched nor control larvae discriminate the larger number of bars. These results are confirmed by the analysis of the proportion of time spent in the two sectors containing the stimuli (one-sample *t*-test; 2 vs. 4 enriched larvae: *t*_9_ = 2.519, *p* = 0.033; control larvae: *t*_9_ = 0.162, *p* = 0.875; 2 vs. 3 enriched larvae: *t*_9_ = −0.391, *p* = 0.705; control larvae: *t*_9_ = 1.273, *p* = 0.235; Figure 6)


**Experiment 3: Preference for enriched environment**


During the first 90 min following introduction in the apparatus, larvae did not show a significant preference for the enriched environment (percentage of time spent in the half of the tank containing the objects: 51.18 ± 8.71%; *t*_(5)_ = 0.661, *p* = 0.538). However, the analysis performed by partitioning the test into 15 min intervals indicate that larvae varied significantly their preference across the test with time spent in the enriched sector increasing significantly as time passed (*F*_(5, 25)_ = 6.181; *p* < 0.001; linear trend: Estimate 0.06 ± 0.02, *t*_(25)_ = 2.980, *p* = 0.006; Figure 7). When the 15 min interval are analysed separately, we found a significant preference for the non-enriched sector in the first 15-min interval (*t*_(5)_ = 2.725, *p* = 0.042); no other interval is significant (all *p*-values > 0.05). Larvae showed a significant preference for the enriched environment in both tests on the following day (morning: 73.32 ± 3.62%; *t*_(5)_ = 15.842, *p* < 0.001; afternoon: 76.76 ± 4.96%; *t*_(5)_ = 13.210, *p* < 0.001; Figure 7). Preference for the enriched environment increased significantly over the 24 h experiment (*F*_(2, 10)_ = 48.484, *p* < 0.001; linear trend: Estimate 0.18 ± 0.02, *t*_(10)_ = 9.073, *p* < 0.001).

## 4. Discussion

Assessing the environmental requirements of zoo, farm, or laboratory animals is often a lengthy and complex process achieved through the accumulation of indirect evidence [47,48,49]. Consequently, recommendations on how to maintain an animal in captivity continually evolve as new knowledge emerges about the needs of that species. This also concerns the welfare of teleost species commonly kept in captivity, which has made significant progress in recent years. As regards fish used in scientific research, the discovery that adult zebrafish exhibit a preference for complex environments and experience stress when isolated from conspecifics or are kept at high densities has led to substantial changes in recommendations for the husbandry of this laboratory species [24,50,51,52,53]. In this study, we provide evidence for the first time that zebrafish may require some environmental complexity from the moment they hatch.

### 4.1. Environmental Complexity and Cognition

In higher vertebrates, some degree of environmental complexity is necessary for the normal development of their nervous system and there is evidence that the characteristics of the growing environment impact sensory and cognitive abilities and influence behaviour in adult teleosts [2,3,5,24]. In the first two experiments of this study, we tested whether environmental complexity has an effect on development of cognition in the first weeks of life. To test these hypotheses, we used two recently developed tests [33,38]. Experiment 1 focused on the capacity to discriminate two areas, whereas Experiment 2 investigated the ability to process numerical quantities.

Overall, the outcomes of these two experiments suggest that the characteristics of the environment during the first two weeks of life indeed influence the development of cognitive abilities in larval zebrafish. However, this effect appears to be comparatively smaller than what is observed in higher vertebrates raised from birth in an extremely impoverished environment [1,3,54].

In Experiment 1, we exploited zebrafish’s ability to choose the larger gap when navigating an obstacle [34]. A previous study showed that this ability is present from the first week of life and is already quite accurate [33]. In our experiment, subjects of the two treatments achieved almost identical scores in each of the four tasks of varying difficulty. As expected from the previous study, we found a significant decrease in performance as the task difficulty increased, indicating that the test was sensitive enough to capture any differences. Together with the Bayesian analysis of the null results, this suggests that that most likely interpretation of this experiment is that environmental complexity has little or no effect on the development of the ability to discriminate continuous quantities.

A recent study [38] found that 7 dpf zebrafish exhibited a preference for an environment with vertical bars and that, when given the choice between two quantities of bars, they tend to stay closer the larger quantity. In that study, larvae proved able to discriminate 1 vs. 4 (numerosity ratio 0.25), 1 vs. 3 (ratio 0.33), 1 vs. 2, and 2 vs. 4 bars (ratio 0.50). This discrimination persisted even when stimuli were controlled for major continuous variables, such as the density of the items or the total area occupied by arrays [38]. In Experiment 2, we used the 0.50 ratio task (2 vs. 4) as in previous study and added a more challenging task with a 0.67 ratio (2 vs. 3). In the latter task, larvae spent the same amount of time near the two stimuli suggesting that they were unable to discriminate this ratio. Conversely, they confirmed the ability to discriminate the 0.50 ratio. When comparing the two treatments, we observed that this ability was present only in the enriched environment group, as larvae from the other group spent a similar amount of time near the two stimuli. Since larvae of the two treatments were equally attracted toward groups of bars but only those from the enrichment treatment selected the larger number, the most likely explanation is that rearing newly hatched zebrafish in an environment devoid of objects hinders the development of numerical skills. Interestingly, in the study cited above, we found that 7 dpf larvae were attracted to vertical bars if previously exposed to bars [38]. Our Experiment 2 would seem to suggest that this preference develops anyway later in life, without the need for previous experience with these stimuli.

One might question why environmental characteristics in the early stages of life exert an impact on numerical abilities rather than on the discrimination of continuous quantities. For several vertebrates, it has been shown that the discrimination of certain object features, such as size, shape, or colour, is present at birth in a form very similar to what is found in adults [55,56,57]. With regard to the ability to estimate an area and to compare the size of two areas, previous studies on zebrafish have found only a small improvement in performance between 7 dpf and adulthood [33,34], suggesting that there is little opportunity for maturation and experience to influence the development of this function.

Conversely, evidence in two species, humans and guppies (*Poecilia reticulata*), indicates that numerical abilities are influenced by experience and maturation [58,59]. In guppies, the capacity to discriminate two numerosities (two groups of conspecifics) is present at birth. Numerical acuity in this task improves over age, but the enhancement is more rapid for guppies reared with a group of conspecifics than for those reared with a single companion. It is conceivable that, in other species too, daily interaction with objects plays a crucial role in developing the ability to enumerate them. The absence of objects in the control condition may have led to an impairment or delay in the development of this ability. Undoubtedly, this aspect warrants further exploration in future research.

### 4.2. Environmental Complexity and Behaviour

Experiments 1 and 2 also allowed the measurement of certain aspects of behaviour, particularly those related to general activity. In general, larvae housed in the enriched environment during the tests exhibited higher levels of motor activity compared to the control group. In the first experiment, this heightened activity is evident in the number of passages between sectors, which was more than twofold greater for the larvae from the enriched treatment. During the second experiment, subjects from the enriched treatment displayed a higher speed and covered a greater distance compared to the controls. One possibility to explain these results is that the type of environment impacted the subjects’ anxiety-like behaviour, which in turn affected their motor activity when placed in an unfamiliar environment. Indeed, two recent studies demonstrated that exposure to environmental complexity reduces the fear of new places or objects in 14 dpf larvae [29,30]. Therefore, it is conceivable that the increased activity observed in our experiments is a consequence of enriched larvae being less cautious when exploring a new space. This interpretation gains further support from the observation that, in Experiment 1, there is a difference between treatments in locomotor activity immediately after introduction to the apparatus, but this difference diminishes after the familiarization phase. However, it is possible that growing in a complex environment has a direct effect on motor activity and exploratory behaviour, as reported for other vertebrates [60,61]. The only study that measured the activity of larval zebrafish in their home-tank reported that 6 dpf larvae that had previously been kept in a group showed higher levels of locomotor activity than those kept alone from birth; the difference in activity was significant during the period of dark, but not during the light period [62]. However, the effects of social enrichment may not be equivalent to those of physical enrichment used in our study. The factors contributing to the differences we observed in locomotor behaviour remain thus unclear and should be addressed in future research.

### 4.3. Preference for Environmental Complexity

Since growing up in a complex environment appears to improve performance for certain cognitive functions and possibly make individuals less fearful, one may wonder whether larvae at birth may already have a preference for more complex environments. In the third experiment, we utilized a classical method for assessing animal requirements and welfare—the preference test [63,64,65]. We placed recently hatched larvae in a container where they could choose between one bare half and one containing several objects of different colours and shapes. Fish were observed soon after being introduced to the apparatus (when they had never experienced any object previously), and on two other instances spaced over the following 24 h.

Overall, our experiment evidenced a clear preference for the enriched half. The preference was not immediately evident soon after introduction. During the first 90 min, subjects spent approximately 50 percent of their time in each half. However, a temporal analysis of their behaviour shows a significant change in preference over time: after an initial avoidance of the enriched sector, larval zebrafish progressively increased the time spent in this sector. One explanation for this result could be traced back to a neophobic behaviour of the larvae. Many species display neophobic tendencies, wherein they instinctively maintain a distance from unfamiliar objects—a behaviour with clear adaptive functions [66,67]. There exists substantial experimental evidence supporting the presence of neophobia in zebrafish, with larvae exhibiting avoidance of unfamiliar objects as early as the first weeks of life [30,68]. The behaviour observed in the initial minutes following introduction might be attributed to the delicate balance between attraction to the environmental complexity and fear of the new objects. Fear of novelty would be most pronounced immediately after introduction but would gradually diminish over time as the novel objects become familiar. Notably, in a previous report, the observed avoidance of new objects tended to fade within approximately 30 min of the initial exposure [69], further supporting the hypothesis that the pattern we observed is due to the initial neophobic response fading out as the environment becomes more familiar to the subjects.

The variation in larval behaviour observed between the first and second day could have alternative explanations. For example, it is possible that 5 dpf larval zebrafish either are unable to distinguish between the two sectors of the tank or exhibit no preference for environmental complexity and that these features only appear the following day, at 6 dpf. The observed temporal fluctuations in the first 90 min after introduction could be due, in this scenario, to some unknown factor. This explanation would fit well with the traditional viewpoint that considers zebrafish having a rather underdeveloped nervous system at hatching and having consequently limited interactions with the physical and social environment [70]. However, as previously mentioned, increasing evidence raises doubts about this view [71,72,73]. Further research is warranted, such as observing larvae at different ages, to elucidate whether the observed differences in behaviour between day 1 and day 2 result from the maturation of the nervous system, a reduction in the neophobic response following familiarization with objects, or some other factors.

The level of preference for the enriched sector is similar in the two observations made the following day. In both instances, the larvae spent more than 70% of their time in the sector containing various objects. This percentage surpasses or matches that observed in comparable research conducted on adult zebrafish (56% in [74]; 67% in [75]). However, making direct comparisons is challenging due to substantial methodological differences between these studies on adults and our study on larvae, with the former being generally extended over longer durations.

In the preceding discussion, we implicitly interpreted the behaviour of larvae as indicative of a preference for structurally complex habitats. However, our simple experiment cannot definitively determine whether this behaviour stems from a preference for complexity itself or is driven by other motivations, such as a tendency to explore novel objects or an aversion to exposed areas. The former explanation seems improbable, as we observed sustained time spent in the enriched compartment even after 24 h, whereas in most vertebrates, exploration of new stimuli typically declines rapidly after initial exposure [76,77,78]. The latter hypothesis may be plausible if open habitats are associated with higher predation risk in nature. However, little is known about the ecology of larval zebrafish and their habitat preferences in nature. Adult zebrafish occupy diverse habitats, including open water and aquatic vegetation and they likely spawn in both environments [79,80].

Resolving this issue will require a series of rigorous controls to differentiate between subtle motivational factors that may interact synergistically or antagonistically. Nevertheless, concerns about the environmental needs and welfare of larval zebrafish persist, whether they seek protection, adequate sensory stimulation, or opportunities for exploration in their environment.

### 4.4. Implications for Welfare

The result of Experiment 3 suggests that we should reconsider our views about the level of maturation of the zebrafish nervous system at birth. If at 6 dpf (possibly at 5 dpf), zebrafish can distinguish between the two halves and make a choice, they likely possess more sophisticated abilities than previously assumed. This evidence aligns with recently gathered observations suggesting that, as early as the first week of life, larval zebrafish exhibit cognitive complexity comparable to that of older larvae and, in some cases, adults. Indeed, a recent study demonstrated that four days after hatching (at 7 dpf), larval zebrafish can discriminate between different numbers of vertical bars, indicating the presence of numerical abilities at this stage [38]. Santacà et al. [32] conducted training sessions using an appetitive conditioning procedure and found that by 9 dpf zebrafish have learnt to discriminated two complex visual patterns and by 12 dpf they could discriminate two objects differing only in colour, shape, or spatial orientation. These learning capacities may indeed be present even earlier as suggested by the presence of aversive conditioning at 7 dpf [31].

A logical consequence of these observations is that, based on the extensive literature indicating that early deprivation often produces dramatic effects on the development of behaviour and cognitive functions in mammals and birds [2,3,5], we would expect that rearing larval zebrafish in an environment completely devoid of stimulation should have significant effects on the development of their neural functions. However, our findings indicate that the impact on cognition and behaviour was mild, suggesting that, unlike higher vertebrates, the quantity and quality of sensory input received during early development might play a minor role in nervous system development in zebrafish. A potential explanation for this phenomenon could be linked to the reproductive mode of egg-laying fish. Unlike mammals and birds, which exhibit parental care and provide a relatively stable environment during early postnatal development, zebrafish lay their eggs in a diverse array of habitats. These habitats may, for example, feature abundant vegetation or be entirely devoid of it, and possess clear or extremely turbid waters [81,82]. Consequently, the trajectory of the nervous system development in zebrafish might be somewhat buffered against the unpredictable conditions prevalent in their diverse breeding environments.

However, this initial study had several limitations in its ability to detect the potential effects of early deprivation on the development of neural functions. First, we have only examined two cognitive functions and it is possible that environmental complexity plays an important role on the development of other cognitive systems. A second possibility is that the environment might exert its effect on the development of cognitive abilities at a later stage in development. Supporting this notion, DePasquale et al. [83] explored differences in behaviour and cognition in juvenile zebrafish by subjecting them to either bare or enriched environments for 52 days starting at 25 days of age. When subjects were trained to find the correct exit out of four alternatives, those from the enrichment treatment demonstrated to be faster at finding the exit of the maze. However, it is worth noting that this measure of memory could be influenced by swimming speed—a parameter we found to be affected by experience with environmental complexity in our study. This highlights the intricacies of the relationship between environmental conditions and cognitive development in zebrafish, warranting further exploration in future research.

A third crucial aspect to consider is that, in Experiments 1 and 2, the environment was impoverished only to a certain extent. Classical experiments conducted on higher vertebrates often involved isolation from conspecifics [1,3]. Due to ethical considerations and the specific focus of our work on assessing the adequacy of current practices for zebrafish husbandry, conspecifics were consistently present in both treatment groups in our study. Even though it is suggested that larvae are not inherently gregarious until they are approximately three weeks old [43], the presence of numerous conspecifics in the tank could generate a significantly higher level of sensory stimulation compared to a completely bare environment [62]. As a result, we cannot disregard the possibility that the effects of rearing in an environment that is both physically and socially impoverished might align more closely with those observed in mammals and birds. This matter undoubtedly warrants future investigation. Understanding the interplay between physical and social aspects of the environment will significantly enhance our understanding of zebrafish neurocognitive development. Furthermore, it will provide valuable insights into the differences in the developmental mechanisms between the nervous system of teleost fish and the larger, more complex nervous systems of warm-blooded vertebrates. This issue is also relevant for the welfare since many research practices involve the complete social and sensory isolation of larval zebrafish, often for long periods of time (e.g., [84,85]).

## 5. Conclusions

Our study suggests that quality of the rearing environment in terms of structural complexity impacts the development of cognition and behaviour of young zebrafish and that recently hatched zebrafish exhibit a preference for specific habitat types and actively seek them. If these results are confirmed, depriving them of these environmental features could potentially impact their welfare. This highlights a previously overlooked concern: even newborn larvae may have specific requirements in terms of the quality of the environment in which they are reared. Furthermore, raising them in an inadequate environment could affect their cognitive development and behaviour, with the consequence that unrecognized confounders could impact the results of the research. Understanding and addressing these issues will contribute significantly to ensuring the well-being of the zebrafish population in captivity and to enhancing the quality and reliability of research involving larval zebrafish.

## Figures and Tables

**Figure 1 animals-14-01031-f001:**
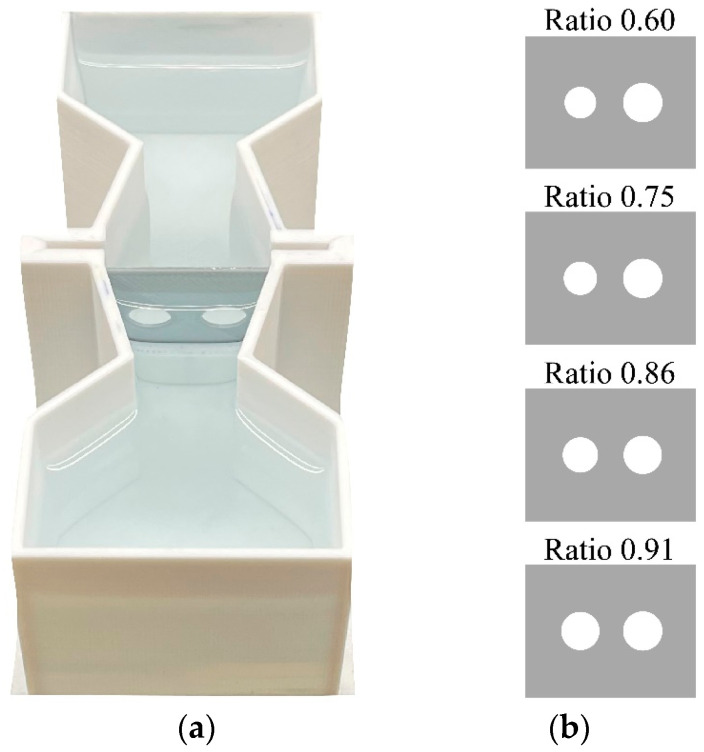
Aerial view of the experimental apparatus (**a**) and panels (**b**) used in Experiment 1. (**a**) The apparatus was divided into a frontal and a posterior sector by a corridor in which a movable test panel was inserted. (**b**) The panels of the test phase presented two holes with four different ratios between the areas: ratio 0.60, ratio 0.75, ratio 0.86 and ratio 0.91.

**Figure 2 animals-14-01031-f002:**
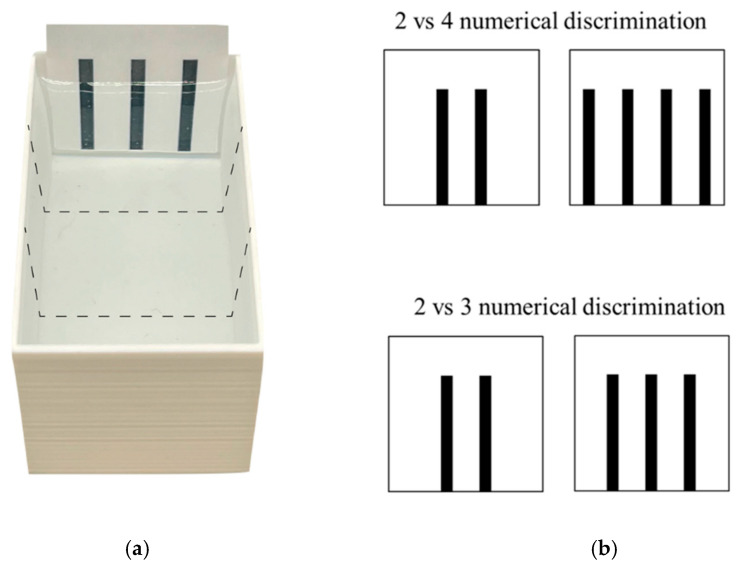
Aerial view of the experimental apparatus (**a**) and stimuli (**b**) used in Experiment 2. (**a**) Larvae were individually observed in a dichotomous-choice test for both 2 vs. 4 and 2 vs. 3 number discrimination. To assess spontaneous preference for the stimuli, we virtually divided the apparatus into three equal sectors (dotted lines): two choice sectors facing the stimuli and one central, no-choice sector. (**b**) Stimuli used in both discriminations were black bars presented on a white background and were matched for density whereas the cumulative surface area changed in parallel with number of bars.

**Figure 3 animals-14-01031-f003:**
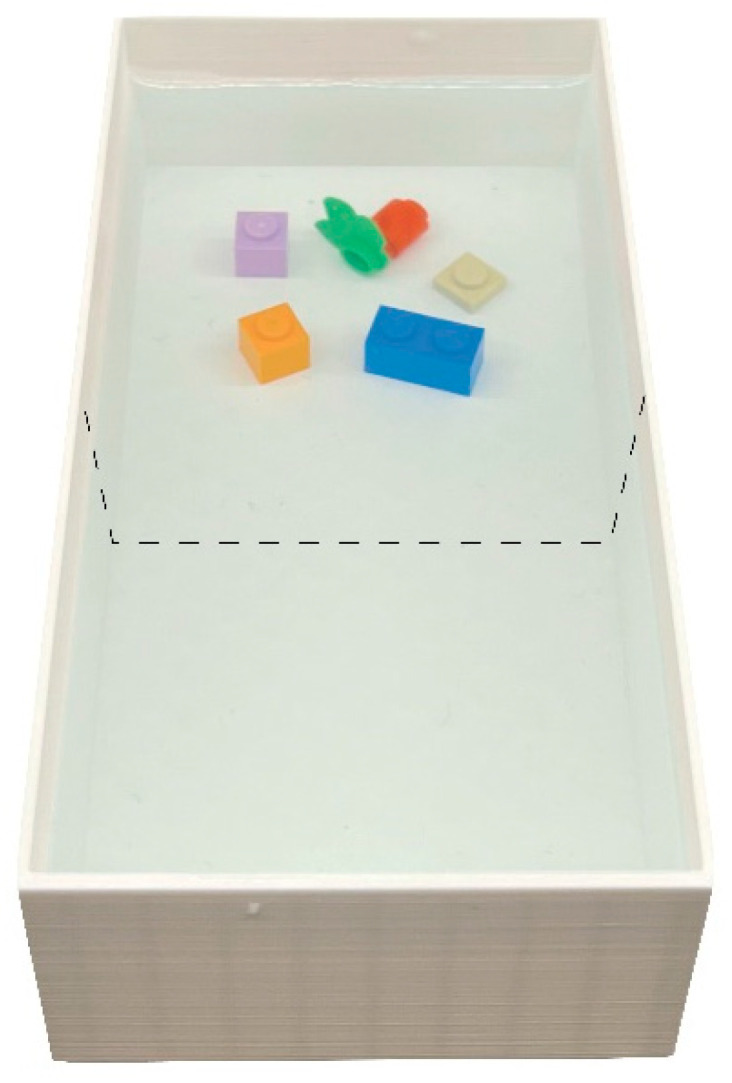
Aerial view of the experimental apparatus of Experiment 3. Twenty larvae were observed in a tank where in one half five Lego^®^ bricks were inserted whereas the other remained empty. To assess preference for the type of environment (enriched vs. barren), we virtually divided the apparatus into two equal sectors (dotted line).

**Figure 4 animals-14-01031-f004:**
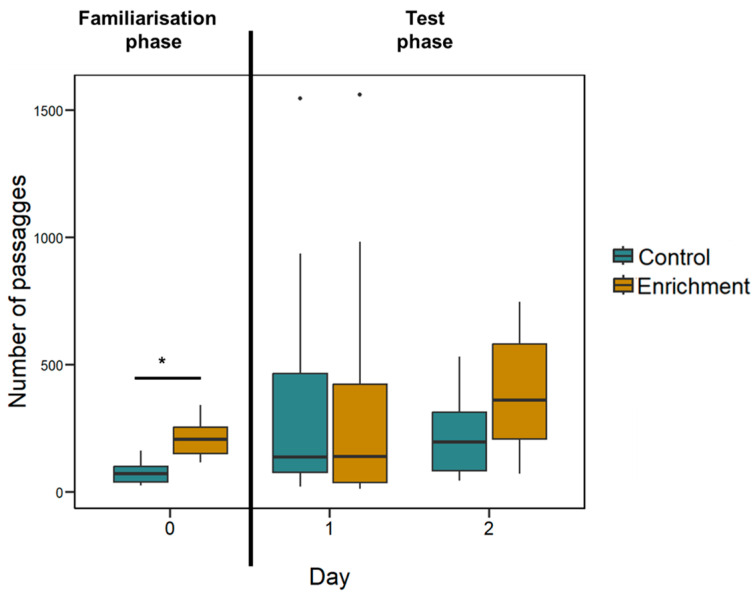
Larvae’s activity recorded in Experiment 1. The *Y*-axis refers to the number of passages through the central hole (familiarisation phase) or the two holes (test phase) in all experimental days. Boxplots represent median, first quartile, third quartile, ranges, and outliers (data points 1.5 interquartile ranges smaller than the first quartile or greater than the third quartile). The asterisk (*) denotes a significant difference between the treatment (*p* < 0.05).

**Figure 5 animals-14-01031-f005:**
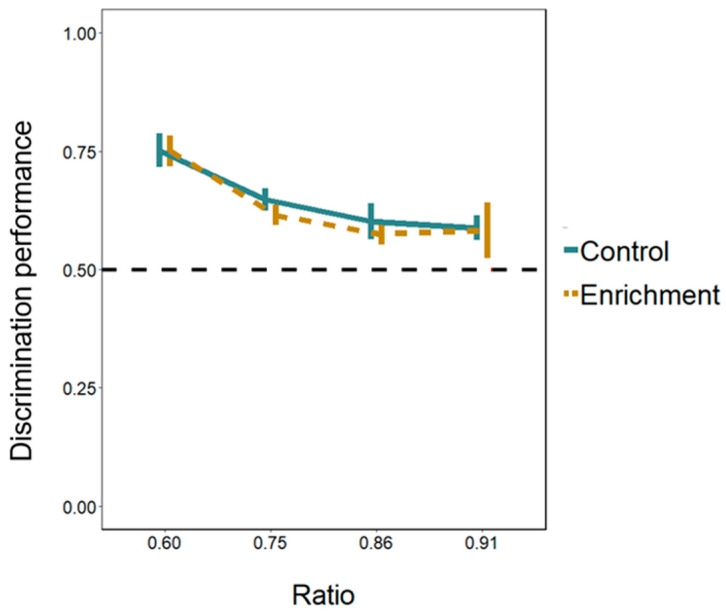
Discrimination of areas in relation holes’ size ratio in Experiment 1. The Y-axis refers to the discrimination performance (proportion of choices for the larger hole) in the four ratios tested (ratio 0.60, ratio 0.75, ratio 0.86, ratio 0.91) for both treatments. Bars represent the standard error. The dotted line shows the chance level (0.5).

**Figure 6 animals-14-01031-f006:**
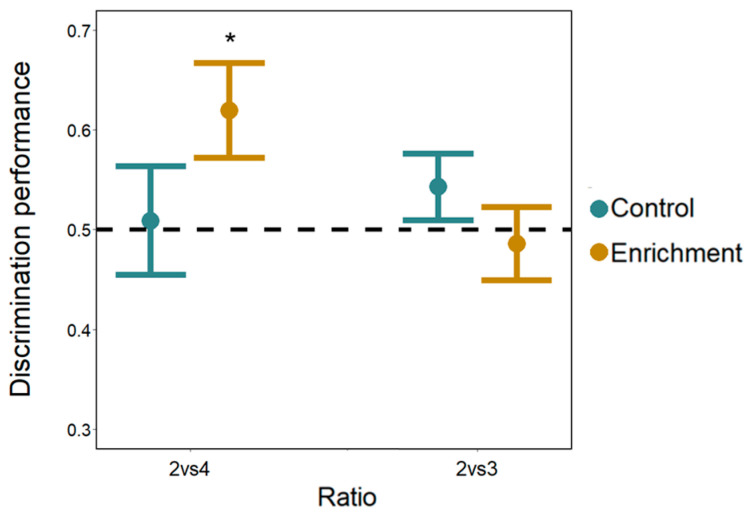
Number discrimination observed in Experiment 2. The Y-axis refers to the proportion of time spent in the choice sectors with the larger number of bars for both number discriminations (2 vs. 4 and 2 vs. 3) and for both treatments (control and enrichment larvae). Bars represent the standard error. Asterisks (*) denote a significant departure from the chance level (0.5) shown by the dotted line.

**Figure 7 animals-14-01031-f007:**
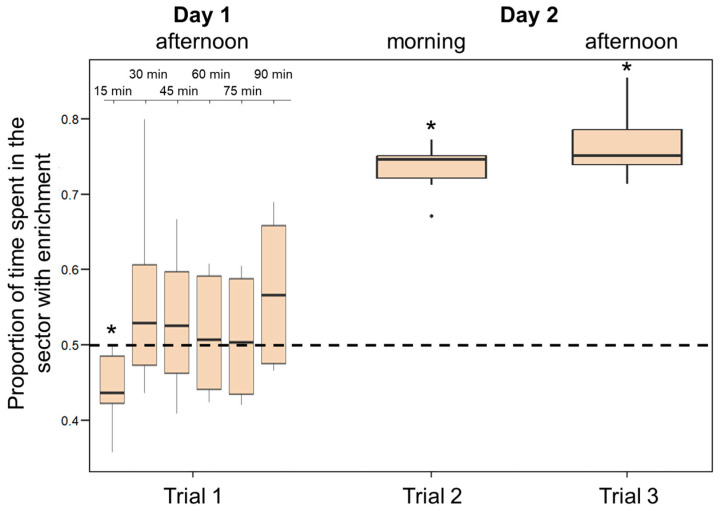
Preference for the enriched sector in Experiment 3. The *Y*-axis refers to the proportion of time that larvae spent in the enriched half of the tank in the three trials. The ninety minutes observation performed soon after their introduction (Trial 1) was analysed in 15 min blocks. Boxplots represent median, first quartile, third quartile, ranges, and outliers (data points 1.5 interquartile ranges smaller than the first quartile or greater than the third quartile). Asterisks (*) denote a significant departure from the chance level (0.5) shown by the dotted line.

## Data Availability

The datasets used for the analyses are provided as Appendix A, further inquiries can be directed to the corresponding author.

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
