# Peer review of "Exploring the Importance of Environmental Complexity for Newly Hatched Zebrafish"

_animals, 2024, doi:10.3390/ani14071031_

Round 1
Reviewer 1 Report
Comments and Suggestions for Authors
This manuscript examines the effect of environmental enrichment on early larval behaviors in the zebrafish. This is an interesting area, as little thought is typically given to the environment of the first week, but the growing evidence that young larvae exhibit complex behaviors and that environment can alter these behaviors makes studies such as this important for both understanding neural development and for considering best husbandry practices.
Overall the manuscript is well arranged and the experiments are clear with understandable data and conclusions. There are numerous typographical and English-language errors throughout the manuscript which should be corrected to improve readability.
A few suggestions for improvements on the manuscript:
1) In Figure 6, the use of Time (log) gives some relative scale of what is happening but is a somewhat complicated measure to parse out exactly what that means. Given that there is a third neutral compartment as well, it might be better to plot % of time in the choice sector which would also allow the neutral time to be inferred.
2) The use of the term numerosity doesn't quite fit the usual definition for that word. For example on line 385, "discriminate the larger numerosity" should just be "...the larger number". In other places numerosity could be just better marked as the number discrimination.
3) In lines 462-464 the authors conclude that rearing larval zebrafish in an unenriched environment might lead to a deterioration of the number discrimination present shortly after hatching. It is not clear why they think the ability is present shortly after hatching as the study they cite here (which is their own study) only found that discrimination when the larvae were reared in a tank with vertical stripes and not in a tank with no visual cues. This detail seems important to include both for the conclusion here and for the general idea of whether these behaviors are innate or learned through environmental interaction. There should be some additional clarity here and in other places where the basis of the behaviors is being discussed, since the experimental basis for this seems to be that the fish needed exposure to stripes to have a preference for more stripes rather than some innate counting ability. The enrichment here expands somwhat that finding, but I don't think it necessarily can lead to the conclusion that there was an innate numerical ability lost without more precise tests.
4) The manuscript discusses one consistent behavioral difference in terms of increased swimming activity in the two tests. It might be helpful to know if these differenes are also seen in the home tanks at the end of the enrichment period or whether it simply relates to behavioral differences in the face of a changed environment for testing. It's probably not necessary to include this, but would be helpful for distinguishing the potential roots of the observed difference.
5) The manuscript discusses in several places attraction to complexity or attraction to greater numbers. It seems that one might need to be careful in inferring too much about the reasons for the positioning, as it could be less about attraction to complexity and more about fear of open space. I'm thinking of other types of behavioral tasks where avoiding open space is often a sign of some anxiety. I think the findings here are important (and I'm going to consider some enrichment of my larval rearing environment!), but I worry a little about whether one can really conclude fine innate cognitive function from these assays (e.g. numerical ability versus hoping to be in an environment that could hide them better). On line 38 of the abstract, as one example, there is a key implication that the choice is of an environment that promotes cognitive development but it's not really clear that that is what the choice is about. Similarly the first line of the Conclusions (Line 615-617) implies that the preference implies a welfare need. It will probably be important to better understand the basis for the preference and the full range of behavioral effects due to the preference and early exposure.
6) The graphs are not consistently sized. This may be an artifact of the pre-production, but in general they could all be reduced in size and use consistent sizing and fonts for aesthetic purposes.
Comments on the Quality of English LanguageThere are a number of places where the writing could be improved. A few examples:
Line 14: Add "have" so it reads "Most studies have looked"
Line 15: "how it impacts during the early stages" needs an object after impacts or just remove the word during.
Line 36: impoverish should be impoverished.
Line 40: "Deepening these aspects" is a little vague.
Line 82: The comma after behaviour could be deleted.
Line 146: The "Each" and "tanks were" don't match signular vs plural
Line 250: no-toxic should be non-toxic
Reviewer 2 Report
Comments and Suggestions for Authors
The manuscript is interesting, concise and easy to be followed. I think it can be accepted for publish in Animals after a few minor revisions.
Line 160, why transferred the larvae at 5 dpf?
Figure 1, and line 198, I suggested the authors mention the life-right position adjustment in the caption of figure 1 or somewhere earlier in the main text.
Figure 2, the cumulative surface area changed in parallel with number of columns
Figure 4, it is ok to present the variables as number of passages, however, I prefer to use the frequency of passages which not affected by experimental observation period
Experiment 2, is the speed mean or median speed? It is interesting, a recent study found that at inter-species level, bolder species exhibited more locomotor for information sampling during shoal selection hence showed high performance of numerical discrimination than shyer species.
Did the authors measured the percentage of time spent moving?
Figure 6 I suggest the authors mark the statistical significance of the difference in the figure.
Reference
Fu SJ, Zhang N, Fan J. Personality and cognition: shoal size discrimination performance is related to boldness and sociability among ten freshwater fish species. Anim Cogn. 2024 Mar 2;27(1):6. doi: 10.1007/s10071-024-01837-x.
Reviewer 3 Report
Comments and Suggestions for Authors
General comments:
This is an excellent, thoughtful, well-executed, and generally well-written study. I have only minor comments regarding wording in several sections of the manuscript.
Specific comments:
L 14: I suggest inserting “have” before “looked”.
L 31: I suggest inserting “under” before “such”.
L 33: Change “are” to “were”.
L 100: For the general reader, please define “dpf” at first use.
L 107: Change “difference” to “differences”.
L 115-117: Please clarify. Larger stripes or more numerous stripes?
L 134-135: What is meant by “natural spawning”? This implies spawning in nature.
L 139: What was the purpose of the methylene blue?
L 172: Change “in” to “of”.
L 190: Change “in” to “on”.
L 424: Insert “of” between “husbandry” and “this”.
L 443: Change “pretty” to “quite”?
L 447: Change “suggest” to suggests”.
L 452: Awkward wording. Please clarify.
L 543: I recommend changing “remains consistent in” to “is consistent with”.
L 569: Change “its” to “their”?
L 572-579: Good point!
Comments on the Quality of English LanguageGood. See my specific comments above.
